# Three New Integration Vectors and Fluorescent Proteins for Use in the Opportunistic Human Pathogen *Streptococcus pneumoniae*

**DOI:** 10.3390/genes10050394

**Published:** 2019-05-22

**Authors:** Lance E. Keller, Anne-Stéphanie Rueff, Jun Kurushima, Jan-Willem Veening

**Affiliations:** Department of Fundamental Microbiology, Faculty of Biology and Medicine, University of Lausanne, Biophore Building, CH-1015 Lausanne, Switzerland; lanceedward.keller@unil.ch (L.E.K.); anne-stephanie.rueff@unil.ch (A.-S.R.); jun.kurushima@unil.ch (J.K.)

**Keywords:** *Streptococcus pneumoniae*, pneumococcal engineering platform, synthetic biology, expression vectors, mTurquoise2, mNeonGreen, mScarlet-I

## Abstract

Here, we describe the creation of three integration vectors, pPEPX, pPEPY and pPEPZ, for use with the opportunistic human pathogen *Streptococcus pneumoniae*. The constructed vectors, named PEP for Pneumococcal Engineering Platform (PEP), employ an IPTG-inducible promoter and BglBrick and BglFusion compatible multiple cloning sites allowing for fast and interchangeable cloning. PEP plasmids replicate in *Escherichia coli* and harbor integration sites that have homology in a large set of pneumococcal strains, including recent clinical isolates. In addition, several options of antibiotic resistance markers are available, even allowing for selection in multidrug resistant clinical isolates. The transformation efficiency of these PEP vectors as well as their ability to be expressed simultaneously was tested. Two of the three PEP vectors share homology of the integration regions with over half of the *S. pneumoniae* genomes examined. Transformation efficiency varied among PEP vectors based on the length of the homology regions, but all were highly transformable and can be integrated simultaneously in strain D39V. Vectors used for pneumococcal cloning are an important tool for researchers for a wide range of uses. The PEP vectors described are of particular use because they have been designed to allow for easy transfer of genes between vectors as well as integrating into transcriptionally silent areas of the chromosome. In addition, we demonstrate the successful production of several new spectrally distinct fluorescent proteins (mTurquoise2, mNeonGreen and mScarlet-I) from the PEP vectors. The PEP vectors and newly described fluorescent proteins will expand the genetic toolbox for pneumococcal researchers and aid future discoveries.

## 1. Introduction

Infectious disease is one of the most common causes of death worldwide and is an object of intense scientific study [1]. A powerful tool into the research of human pathogens is the ability to manipulate the pathogen in a desired manner to observe the outcome. These manipulations can take many forms; gene deletions, genetic complementation, gene mutations, or controlled expression. To aid in alteration of a target genome, numerous methods have been developed over the years with varying success depending on the targeted pathogen [2]. Tool development in model organisms tend to be over represented compared to other species, with bacterial human pathogens having fewer options. While the study of model organisms has led to great advances in understanding the nature of diverse bacterial functions, they are not useful for pathogen specific processes. In an attempt to ameliorate this deficit, we set out to develop several cloning vectors useful for genetic manipulation in *Streptococcus pneumoniae*, even in recent clinically relevant isolates.

The bacterium *S. pneumoniae* is an important opportunistic human pathogen that is responsible for over a million deaths per annum [3]. *S. pneumoniae* has been extensively studied in the past decade and several useful tools for genetic manipulation have been created that allow for stable single copy integration at the genome. However, there is still a lack of robust suicide vectors that do not disrupt native expression of important genes that can be combined to insert several genes in the same strain or can be applied to multi-drug resistant clinical isolates. One of the most commonly used ectopic integration site in *S. pneumoniae* is the *bgaA* locus [4,5,6,7], which disrupts this β-galactosidase, which might have an important function during colonization of human epithelial cells [8,9,10]. The only neutral integration site for which integration plasmids have been developed is the CEP locus (chromosomal expression platform), which is a transcriptionally silent site within a truncated IS 1167 element downstream of the well-studied *ami* operon [11,12]. However, this loci is not very well conserved among pneumococcal strains (see Results below).

Utilizing extensive RNA-seq data for *S. pneumoniae* in multiple conditions [13], transcriptionally silent regions were identified for potential chromosomal locations to use as recombination sites. Using the pneumococcal engineering platform (PEP) vector as a guide [12], novel integration vectors were developed for use at these regions. In this study, we have developed and tested three new PEP vectors, pPEPX, pPEPY, and pPEPZ, for their ability to be transformed into *S. pneumoniae*, compatibility for co-expression, and frequency of occurrence in pneumococcal isolates. In addition, three spectrally distinct and intrinsically bright fluorescent proteins, mTurquoise2, mNeonGreen, and mScarlet-I, were added to the fluorescence protein toolbox now available for *S. pneumoniae*.

## 2. Materials and Methods

### 2.1. Growth Conditions

All pneumococcal strains are derived from the parent strain D39V [14] and were grown in the semi-synthetic media C + Y formulation 2018 [15] under selection conditions where applicable, see Table 1. Standard pneumococcal transformation [16] protocols using synthetic CSP were used for strain construction with selection on Columbia agar supplemented with 2% sheep blood and appropriate antibiotics. Plasmids were cloned and maintained in *Escherichia coli* strains XL1-Blue or Stbl3 with LB media and selected on LB agar plates with appropriate antibiotics (Table 1). Transformation into *E. coli* was done using rubidium chloride chemically competent cells with standard procedures. Induction of all strains containing an isopropyl β-D-1-thiogalactopyranoside (IPTG)-inducible promoter were activated with 1 mM of IPTG for 1 h before visualization by microscopy.

### 2.2. Plasmid Construction

RNA-seq data available at https://veeninglab.com/pneumoexpress [13] and gene annotation data at https://veeninglab.com/pneumobrowse [14] was used for selection of transcriptionally silent regions.

pPEPX. This vector is a modified smaller version of pPEP1 [12] lacking the chloramphenicol resistance marker and containing the high copy number origin of replication from pUC18, but still integrates at the neutral CEP locus near *amiF* [11]. The pUC18 origin of replication was amplified from pLA01 [17] using primers pUC18-F-KpnI and pUC18-R-NotI. The amplified fragment containing pUC18 origin and pPEP1 were digested with *Kpn*I and *Not*I (NEB, Ipswich, MA, USA) for 1 h at 37 °C followed by overnight RT ligation with T4 DNA ligase (ThermoScientific, Waltham, MA, USA). The resulting plasmid was then digested with *Kpn*I and *Nco*I (NEB, Ipswich, MA, USA) for 1 h at 37 °C and ligated with hybridized primers NcoI-KpnI-link-F and NcoI-KpnI-link-R that were similarly digested. The IPTG inducible promoter was added to the promoterless pPEPX plasmid by digestion of pPEP8PL8-2 [18] with *Eco*RI and *Bgl*II and ligating appropriate fragment to complementary digested pPEPX.

pPEPY-Creation of plasmid was completed through Gibson Assembly (NEB, Ipswich, MA, USA) using a minimum of 20 bp overlapping regions of five DNA fragments following manufacturers protocol. Integration regions were amplified from D39V genomic DNA using primer pairs LK042/LK043 and LK046/LK047. Origin of replication and multiple cloning site, which contains transcriptional terminator within the site, was amplified from pPEPX vector using primer pairs LK044/LK045 and LK040/LK041 respectively. Kanamycin resistance cassette was amplified from plasmid pDG783 [19] with primer pair LK033/LK034 that included addition of the P_F6_ promoter [12] for kanamycin expression. Three terminators were added after the multiple cloning site to this vector through Gibson assembly of terminators from plasmid pJWV100 [7] (OVL68/69) and the pre-pPEPY plasmid (OVL66/67), creating pPEPY. The IPTG inducible promoter (P_lac_) [20] was added through restriction digestion of P_lac_ amplified from FtsK-mNeonGreen (van Raaphorst and Veening, unpublished) with primer pair OVL64/65, creating pPEPY-P_lac_. Plasmid and DNA was digested with *Nhe*I and *Eco*RI followed by ligation and transformation into *E. coli* strain XL1-Blue. The pPEPY plasmid integrates at chromosomal integration locus (CIL) thereby interrupting the non-essential pseudogene *spv_2165*.

pPEPZ-Creation of an integration vector of *S. pneumoniae* at the locus *spv_2417* (aka *spd_1735*) was completed through Golden Gate cloning. Primers were designed with Benchling to amplify four different DNA fragments. Primer pairs OVL681/682 and OVL683/684 were used to amplify the downstream and upstream homologous region, respectively, for integration into *S. pneumoniae* by using chromosomal DNA of D39V as template. Primers OVL685/686 were used to amplify the pUC18 origin of replication of from pPEPX vector and primers OVL687/688 were used to amplify a fragment containing the promoter P_lac_, the multiple cloning site, and the spectinomycin marker from the pPEP1-P_lac_ vector [12]. *Dpn*I treatment was performed for 15 min at 37 °C and the four different amplified parts were purified and then digested with *Bsm*BI (NEB, Ipswich, MA, USA) enzyme at 55 °C for 1.5 h. This reaction was purified and ligated at RT for 1.5 h then transformed into *E. coli* XL1-Blue. Digestion control followed by sequencing was performed to confirm the construct. The pPEPZ vector integrates at the PEPZ Integration Position (ZIP) locus upstream of *spv_1736* thereby interrupting the non-essential pseudogene *spv_2417*.

To generate the pPEPZ variants with gentamycin (*gent*) resistance, a Golden Gate strategy was employed. Primer pair OVL1066/1067 was used to amplify pPEPZ-P_lac_ and primers pair OVL1068/1069 was used to amplify the *gent* gene from D39V containing the *lacI* gene at the *prs1* locus as template [20]. The two PCR products were purified, digested with *Bbs*I-HF (NEB, Ipswich, MA, USA) at 37 °C for 1 hr, and ligated at RT for 1 h. Ligation mixture was used to transform *E. coli* strain Stbl3, sequencing was performed to confirm the construct called pASR102.

pPEPZ containing *bla*- First, β-lactamase gene (*bla*) was cloned into the pPEPZ-Plac (*spec*) vector in addition to the spectinomycin (*spec*) marker. Primer pairs OVL1190/1191 was used to amplified the *bla* gene from pASR113 as template (a pLA18 derivative, Rueff and Veening unpublished, [17]). The purified PCR fragment and pPEPZ-P_lac_ vector were digested for 1.5 h at 37 °C with *Not*I (NEB, Ipswich, MA, USA) to introduce the *bla* gene at the unique NotI site of pPEPZ-P_lac_ (*spec*), between the origin of replication of pUC18 and the upstream homologous region. Fragments were ligated at RT for 2 h and transformed into *E. coli* strain Stbl3 to generate intermediate plasmid pASR101. Second, this plasmid was used for resistance marker exchange of the spectinomycin marker with an erythromycin (*erm*) marker by Golden Gate cloning. For that, pASR101 was amplified by PCR with primers OVL1066 and OVL1067, and the *erm* gene was amplified with primers OVL1070 and OVL1071 using strain XL39 (Liu and Veening unpublished) as template. The two PCR products were treated with *Dpn*I (NEB, Ipswich, MA, USA) for 30 min at 37 °C, followed by purification and then digested with *Bbs*I-HF (NEB, Ipswich, MA, USA) enzyme for 1 h at 37 °C. Reaction was purified and ligated for 30 min at room temperature and used to transform *E. coli* strain Stbl3. Sequencing was performed to confirm the construct called pASR103. Sequencing revealed an insertion between the *erm* marker and the P_lac_ region. This insertion of 777 bp is from *E. coli* chromosomal DNA and does not affect the functioning of the plasmid, as the P_lac_ promoter is still well insulated by the transcriptional terminators, but does increase plasmid size.

Another version of the pPEPZ plasmid has been generated by replacing the *erm* marker with a trimethoprim (*tmp*) antibiotic marker. The plasmid was obtained by Golden Gate cloning as follows: primers OVL1696/1697 were used to amplify pASR103 and then *Dpn*I (NEB, Ipswich, MA, USA) treated for 1 h at 37 °C. Primers OVL1698/1699 were used to amplify the *tmp* gene from pneumococcal strain AM39 [21] as template. Thereafter, the two PCR products were purified and then digested with *Aar*I (NEB, Ipswich, MA, USA) enzyme at 37 °C for 5 h. After purification, the two parts were ligated at room temperature (RT) for 2 h. The ligation mixture was used to transform *E. coli* strain Stbl3, and subsequent sequencing was performed to confirm the construct called pASR130. This plasmid also contains the *E. coli* chromosomal DNA insertion of 777 bp in between *tmp* and the P_lac_ promoter region, since it is based on the pASR103 plasmid.

All plasmids were verified by sequencing and deposited at Addgene (plasmid# 122631-122641).

Three additional pPEPY-P_lac_ plasmids were created for carboxy(C)-terminal fusions of three spectrally distinct fluorescent proteins, mTurquoise2 [22], mNeonGreen [23], and mScarlet-I [24]. The plasmid pPEPY-P_lac_-link-mNeonGreen was created through amplification of *mNeonGreen* containing an 11 amino acid C-terminal linker with primers link-mNG-F and link-mNG-R from plasmid pRRneon, which contains a *S. pneumoniae* codon optimized version of *mNeonGreen* (Raaphorst and Veening, unpublished). The resulting fragment was digested with *Bam*HI and *Xho*I (NEB, Ipswich, MA, USA) for 1 h at 37 °C followed by overnight RT ligation with similarly digested pPEPY-P_lac_ plasmid and T4 DNA ligase (ThermoScientific, Waltham, MA, USA). The resulting plasmid was used for the template for pPEPY-P_lac_-link-mTurquoise2 using primers OVL914/915. *mTurquoise2* was amplified with OVL916/917 using Addgene plasmid #54844 (mTurquoise2-pBAD) as template. Subsequent DNA fragments were digested in a single reaction with *Bsm*BI (NEB, Ipswich, MA, USA) for 1 h at 55 °C followed by overnight ligation at RT with T4 DNA ligase (ThermoScientific, Waltham, MA, USA). The *mScarlet* gene was obtained through *S. pneumoniae* codon optimized gene synthesis by IDT (Integrated DNA Technologies) followed by cloning into pPEPY-P_lac_. The *mScarlet-I* variant was made through round PCR of pPEPY-P_lac_-mScarlet with primers OVL419/420 introducing a T73I point mutation to the protein. Subsequent amplification was digested with *Dpn*I (NEB, Ipswich, MA, USA) for 1 h at 37 °C followed by treatment with ClonExpress (Vazyme, Nanjing, China) following manufacturers protocol. The *mScarlet-I* variant was amplified from the resulting plasmid with OVL908/909 and combined with linearized pPEPY-P_lac_-Link (OVL906/907) and treated as for pPEPY-P_lac_-link-mTurquoise2 creation. All ligation products were subsequently cloned into XL1-Blue and verified by sequencing. Addgene is pending accession number.

Gene fusions for ectopic integration at various PEP loci were created as follows. The fusion containing ComEA and mTurquoise2 was created through *comEA* amplification from D39V gDNA with primers OVL576/615. Amplification was digested with *Eco*RI and *Bam*HI (NEB, Ipswich, MA, USA) for 1 h at 37 °C and ligated with similarly digested pPEPY-P_lac_-Link-mTurquoise2 followed by overnight ligation at RT with T4 DNA ligase (ThermoScientific, Waltham, MA, USA). The HlpA and mNeonGreen fusion in pASR103, pPEPZ plasmid with erythromycin selection in *S. pneumoniae*, was created by *hlpA-mNeonGreen* amplification from strain VL1978 (Veening, unpublished) with primers OVL1569/1570. The resulting fragment and pASR103 were digested with *Bgl*II and *Xho*I (NEB, Ipswich, MA, USA) for 1 h at 37 °C followed by overnight ligation at RT with T4 DNA ligase. CEP locus integration of *spv_1159-mScarlet-I* was created through amplification of VL1785 gDNA, which contains CEP integration with a *spv_1159-sfGFP* fusion, (Kurushima and Veening unpublished) with primer pairs OVL037/869 and OVL040/870. The fragment containing *mScarlet-I* was amplified from plasmid pPEPY-P_lac_-Link-mScarlet-I with primer pair OVL871/872. The resulting three fragments were digested together with *Bsm*BI (NEB, Ipswich, MA, USA) for 1 h at 55 °C followed by overnight ligation at RT with T4 DNA ligase. All oligos are listed in Appendix A.

### 2.3. Strain Construction

Parental strain D39V was transformed with a constitutively expressed LacI under gentamycin selection, downstream of the *prs1* locus [20], leading to strain VL236. This strain was used for all future transformations, with the exception of plasmid pASR102, which was transformed into D39V, due to gentamycin selection of the plasmid. Standard pneumococcal transformation procedure was used to integrate all plasmids independently with appropriate selection [16]. For the strain expressing all plasmids (strain VL2706), each ligation product was sequentially cloned with verification of appropriate integration between each transformation. Double crossover by homologous recombination was tested after transformation of plasmids into D39V chromosome using primer pairs located outside the integration regions. For integration of pPEPX primer pair OVL873/OVL874 was used, for pPEPX integration primer pair LK352/LK353 was used and for pPEPZ integration primer pair OVL1004/OVL1005 was used.

### 2.4. Database BLAST

Basic Local Alignment Search Tool (BLAST) was done using NCBI blast-2.7.1+ against whole genomes with a minimum assembly at the contig level. All available *S. pneumoniae* sequences were downloaded from the NCBI database and combined with the Maela database (5029 strains). Genomes of closely related species *S. pseudopneumoniae* (43 strains) and *S. tigurinus* (19 strains) as well as other streptococcal species *S. mutans* (190 strains), *S. equi* (259 strains), *S. pyogenes* (449 strains)*,* and *S. suis* (1261 strains) were downloaded from the NCBI database. Integration regions were used as the query sequences against BLAST databases. Results were filtered to include matches with over 95% sequence homology as well as containing at least 90% of the queried DNA. Duplicate matches were removed and results of both integration regions per plasmid were combined. A positive match was only identified if the combined filtered results included two hits per strain, one for upstream integration region and one for downstream integration region. Genetic circular map and comparative alignments were generated using GenomeMatcher and arcWithColor software with modification in graphic representation [25].

### 2.5. Transformation Assay

Wild type *S. pneumoniae* strain D39V was grown to OD_600_ 0.1, washed and suspended in pre-warmed non-acidic C+Y media. Competence was activated with 50 ng/mL of CSP-1 at 37 °C for 10 min and 45 µL was added to a 96 well plate. Five microliters of DNA was added at various concentrations as indicated and incubated for 120 min at 37 °C. Samples were serially diluted and plated on non-selective blood agar (BA) and BA with the appropriate antibiotics. Transformation efficiency was calculated by taking the difference between the CFU of the non-selective and the selective BA. For transformation of strains other than D39V, the above protocol was followed with the addition of 50 ng/mL of CSP-2 for initial competence activation due to variations in ComD specificity.

### 2.6. Microscopy

Fluorescence microscopy was performed on a Leica DMi8 through a 100x phase contrast objective (NA 1.40) with a SOLA Light Engine (lumencor) light source. Light was filtered through external excitation filters 430/24 nm (Chroma ET430/24x), 470/40 nm (Chroma ET470/40x), and 545/25 nm (Chroma ET545/25x) for visualization of mTurquoise2, mNeonGreen, or mScarlet-I, respectively. For mTurquoise2, light passed through a cube (Leica 11536022) containing a polychroic mirror 455/520/595 nm while mNeonGreen and mScarlet-I used a cube (Leica 11536022) with a GFP/RFP polychroic mirror (498/564 nm). External emission filters used were from Chroma and ET470/24m, ET520/40m and ET605/70m were used for mTurquoise2, mNeonGreen and mScarlet-I, respectively. Exposure times of 700 ms, 200 ms, and 500 ms with 100% of light from SOLA Light Engine were performed for mTurquoise2, mNeonGreen, and mScarlet-I fusions respectively. Note that we also routinely image mNeonGreen through YFP filters (Chroma ET500/20x for excitation, Chroma ET535/30m for emission with a CFP/YFP/mCherry (455/520/595) polychroic mirror) in case there is signal bleed-through from the mTurquoise2 label in the GFP channel. Images were captured using LasX software (Leica) and exported to ImageJ [26] for final preparation.

## 3. Results

### 3.1. Design and Construction of pPEPX, pPEPY and pPEPZ

As seen in Figure 1, the three pneumococcal engineering platform plasmids created have compatible cloning sites following the BglBrick and BglFusion principle [12,27], to facilitate easy exchange between the plasmids. The exception is in the pPEPY vector with the *Eco*RI site located downstream of the P_lac_ promoter as opposed to upstream as in the other two plasmids. The integration regions vary for each vector, but are in transcriptionally silent areas of the genome as determined by RNA-seq data [13]. The pPEPX vector integrates about 295° around the chromosome between the *treR* (*spv_1665*) and *amiF* (*spv_1666*) genes (Figure 2A,B). The pPEPY vector integrates at 75° around the chromosome between *spv*_*2165* and *spv*_*2166*, both encoding hypothetical proteins, (Figure 2A,B). The last constructed vector, pPEPZ, integrates at 304° around the chromosome, upstream of *spv*_*1736*, which encodes a degenerative hypothetical protein (Figure 2A,B). Transcriptional terminators flank the region containing the P_lac_ promoter and multiple cloning site (MCS) in all vectors to ensure that the inserted DNA is not influenced by transcriptional events happening elsewhere on the chromosome. The origin of replication used for all plasmids is a high copy number *E. coli* origin derived from pUC18 that is not recognized by *S. pneumoniae*. All selective markers are expressed in both *E. coli* and *S. pneumoniae.*

Various antibiotic markers were created for pPEPZ to facilitate integration of multiple PEP vectors simultaneously, as well as providing multiple options depending on the need of the researcher. As pPEPZ has a large pneumococcal host range (see below), we also constructed pPEPZ variants with antibiotic resistance alleles less commonly found in clinical isolates such as trimethoprim and gentamycin (Figure 2). No PEP vectors contain the tetracycline resistance marker for selection, allowing strains to utilize PEP vectors in combination with the popular pPP1/pJWV/pKB *bgaA-*based integration vectors [5,6,7]. Table 1 summarizes vectors and available resistance markers.

Vector construction was done through various cloning techniques, but common motifs were kept for all PEP vectors. These include the same origin of replication for a high copy number in *E. coli*, similar multiple cloning sites to facilitate rapid exchange between plasmids, an IPTG inducible promoter with tight expression control, and multiple antibiotic selection markers for use in almost any background. Expression without an inducible promoter can be done using pPEPY or standard cloning can be used to exchange the promoter P_lac_ with any desired promoter. A list of available vectors and the selectable markers can be found in Table 1. Note that to make use of the P_lac_ IPTG inducible promoter (Sorg et al. in preparation), the pneumococcal strain being transformed also needs to encode the LacI repressor from *E. coli* otherwise the promoter will be constitutively expressed. We noticed significant vector stability when leaving out the gene encoding the *lacI* repressor from the plasmid. Therefore, we used a separate, ectopic integration of *lacI* [20]. This *lacI* construct can be obtained via Addgene (catalog #85589) and is integrated at the *prs1* locus by selecting for gentamycin [20].

### 3.2. High Transformation Efficiency of pPEPX, pPEPY and pPEPZ

Transformation efficiency of these three vectors was determined in *S. pneumoniae* strain D39V (Figure 3). Multiple concentrations of DNA were tested and the amounts used were normalized to molar concentrations to remove concentration effects due to size differences in the plasmids. It was found that pPEPZ had higher transformation efficiencies than either pPEPX or pPEPY at all DNA concentrations tested (0.064, 0.32, 1.6, 8.0, 40.0, 200.0, 1000.0 nM), with a maximum transformation efficiency of approximately 10^−3^ compared to 10^−4^. This is most likely due to the longer regions of homology present on pPEPZ (1 kb) compared to the homologous integration regions present on pPEPX (500 bps) and pPEPY (500 bps). For all vectors tested there was a two-log increase in transformation efficiency from the lowest concentration (0.064 nM) of DNA added to the highest concentration (1000 nM). A rapid increase in transformation efficiency was observed upon increasing DNA concentrations, but gradually leveled off at a concentration of 40 nM of DNA for all PEP vectors. These data are completely in line with previously described transformation efficiencies in *S. pneumoniae* that demonstrated a similar relation between length of the homology regions and DNA concentration [28,29]. Frequency of double crossover recombination was also determined for each vector by amplification of integration regions with oligos located outside of the integration regions included in the PEP vectors. This is an important consideration to verify correct integration and ensure removal or the vectors necessary for maintenance in *E. coli*, i.e., origin of replication and ampicillin resistance marker. Out of 10 colonies screened for double crossover recombination of each vector in D39V we found a 100% efficiency at integration at the correct locus by double crossover, even without linearizing the vector prior to transformation (Appendix A). In summary, all described vectors can be efficiently transformed through double crossover events into strain D39V and acquisition of one vector does not exclude integration of other PEP vectors at available integration loci.

### 3.3. Simultaneous Integration and Expression of Three New Fluorescent Proteins for S. pneumoniae from the pPEPX, pPEPY and pPEPZ Loci

To demonstrate the feasibility to clone and construct a strain that expresses heterologous proteins from all three integration platforms simultaneously, we utilized several recent spectrally distinct fluorescent proteins not yet widely used in the pneumococcal research field, namely mTurquoise2 [22], mNeonGreen [23], and mScarlet-I [24]. To make sure these proteins do not demonstrate any spectral overlap under our experimental conditions, we fused them to genes encoding proteins with a distinct cellular localization. We cloned the genes encoding SPV_1159-mScarlet-I in pPEPX, ComEA-mTurquoise2 in pPEPY and HlpA-mNeonGreen in pPEPZ and then transformed them into strain D39V in serial (see materials and methods. Note that we directly transformed the ligation mixtures to *S. pneumoniae* without an intermediate *E. coli* clone [30].

Expression of all fusion proteins in constructed strain VL2706 was induced by addition of 1 mM IPTG for 1 h before visualization by fluorescence microscopy. We selected ComEA as this is a protein involved in DNA uptake and was previously shown to demonstrate a septal localization [31]. HlpA (HU) was chosen as it binds to the nucleoid and is a good marker for the chromosome [32,33]. SPV_1159 was selected as it is a predicted small membrane protein with two transmembrane domains with both the N- and C-terminus inside the cytoplasm and was previously shown to demonstrate homogenous membrane localization in our laboratory (Gallay and Veening, unpublished work). As shown in Figure 4, cellular morphology of this triple labelled strain is indistinguishable from wild type and we obtained clear, distinguishable fluorescent signals for all three fusion proteins upon IPTG induction. In addition, all fusions demonstrated their predicted localization. In addition, growth curves of parent strain D39V and VL2706 were indistinguishable after 12 h of growth indicating that integration and the expression of genes from the integration sites did not affect growth (Appendix A). This demonstrates that stable integration and simultaneous expression of heterologous proteins from pPEPX, pPEPY, and pPEPZ is possible, and provides a new set of fluorescent proteins for use in *S. pneumoniae*.

### 3.4. Broad Pneumococcal Strain Range for pPEPY and pPEPZ

All vectors were designed and based on the genome and transcription data of *S. pneumoniae* strain D39V [13,14,34]. To determine if the constructed vectors can be used in multiple strains of *S. pneumoniae* as well as closely related species, whole genome sequences were queried for the integrative regions of the PEP vectors. All genomes from NCBI at contig level or above were downloaded for *S. pneumoniae* (2875 strains), *S. pseudopneumoniae* (38 strains), *S. tigurinus* (19 strains), *S. mutans* (190 strains), *S. equi* (259 strains), *S. pyogenes* (449 strains)*,* and *S. suis* (1261 strains). Additional genomes for *S. pneumoniae* were obtained from the Maela database [35] (3085 strains) for a total of 5960 pneumococcal genomes. Only 5029 pneumococcal genomes were tested due to duplication of some sequences between the two databases (Figure 5). A positive hit for the strain was determined when both integrative regions with a homology of at least 95% were present and had at least 90% of the total integration region. The pPEPX vector passed these requirements for 253 (5.03%) of *S. pneumoniae* strains tested. The pPEPY vector had 3237 (64.37%) positive hits for *S. pneumoniae*. Similar results were found for the pPEPZ vector with 3229 (64.21%) hits for *S. pneumoniae* (Figure 5). All pneumococcal genomes queried can integrate at least one of the PEP vectors available based on sequence homology. For all other streptococcal species tested there were no strains that passed our criteria, but significant homology was found in both *S. pseudopneumoniae* and *S. tigurinus,* which are more closely related to *S. pneumoniae* than other strains tested [36]. Interestingly, the CEP locus for pPEPX integration was the most common locus in other species with some positive hits also found in *S. suis*. Appendix A details which strains contained positive hits for the integration regions of the PEP vectors.

Testing for transformation efficiency in numerous strains and species with multiple vectors is tedious and time consuming and testing a small portion of collected strains will not be as beneficial. Although competence for transformation is highly conserved in *S. pneumoniae* and most strains can be transformed in the laboratory, significant differences in transformation efficiencies of different strains are observed [37]. Through using sequencing data, it is possible to test thousands of strains for the presence of the integrative regions of the constructed plasmids. While this is not conclusive evidence for the ability of the PEP vectors to be transformable into queried strains, it is indicative that they are transformable. Through using stringent cut off conditions of at least 95% homology to the integration regions and the queried strains having 90% or more of the homology region, the chances of successful genomic integration are high. Appendix A list all pneumococcal strains that passed our criteria and indicates possible transformability with the PEP vectors.

### 3.5. Efficient Cloning in Multidrug Resistant Clinical Isolates Using PEP Vectors

Antibiotic resistance is on the rise and several clinically relevant strains are now commonly found worldwide, such as many PMEN (Pneumococcal Molecular Epidemiology Network) strains [38]. To determine the feasibility of using these plasmids in multi-drug resistant clinical isolates we tested the ability of PMEN strains 1, 3, 14, 18, 28 and TIGR4 to integrate PEP plasmids. These are serotype 23F, 9V, 19F, 14, 1, and 4, respectively, and have different antibiotic susceptibilities. Table 2 summarizes the strains, susceptibility, and transformability based on sequence and the experiment. All strains passed our transformability requirements based on genome sequence for at least one PEP vector. Appendix A demonstrate the homology of the integration regions of the PEP plasmids for several clinically relevant and antibiotic resistant strains. Experimentally determined transformability with the PEP vectors verified the feasibility of using a strains’ genome sequence to indicate the ability to integrate a PEP vector. Even when we did not find a high level of homology, we could still transform strains PMEN3 and PMEN14 with plasmids pPEPX and pPEPY. The exception was for the pPEPY vector, which did not yield any transformants with PMEN18 or TIGR4 despite both having high sequence homology to the integration region.

## 4. Discussion

Here, we describe the creation and testing of three integration vectors for use in *Streptococcus pneumoniae*. These vectors can be used for a wide range of applications in the pneumococcal field; complementation of gene deletions, labeling and expression of genes of interest, and/or expressing non-native proteins. The ability to efficiently transform and express any gene of interest stably from the chromosome as a single copy from a neutral locus is an essential tool for pneumococcal research. The PEP vectors described here can potentially be used in a large portion of pneumococcal strains and be of benefit to many researchers in the pneumococcal field. All described vectors are made available through Addgene (catalog numbers pending). It is our hope that making these vectors available will facilitate the research of other groups expanding our knowledge base of *S. pneumoniae*.

## Figures and Tables

**Figure 1 genes-10-00394-f001:**
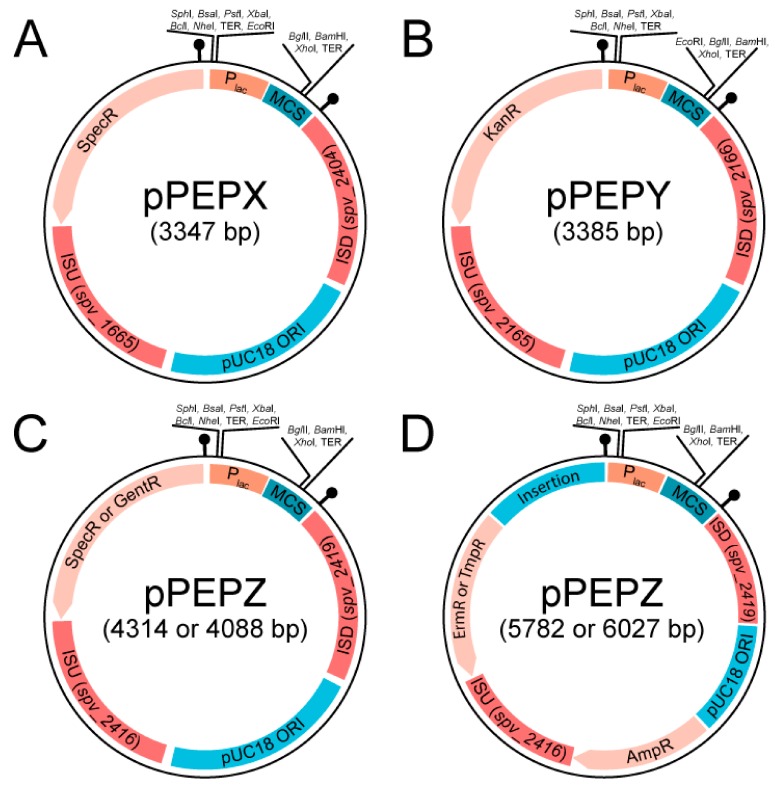
Diagram of general layout for various constructed PEP vectors. All PEP vectors have the same IPTG inducible promoter as well as the same pUC18 origin of replication which is permissive in *E. coli*, but not in *S. pneumoniae*. All vectors have a multiple cloning site featuring the BglBrick cloning method and integration regions that are transcriptionally silent based on RNA-seq data. Each promoter and MCS is flanked by strong terminators. (**A**) The pPEPX vector has a spectinomycin resistance marker and integration regions of approximately 500 bp in length and integrates at 295.4° around the D39V chromosome near *amiF* (*spv_1667*). (**B**) The pPEPY vector has a kanamycin resistance marker and integration regions of approximately 500 bp in length and integrates at 74.6° around the D39V chromosome near *spv_0422*. (**C**) Two pPEPZ vectors have either a spectinomycin or gentamycin resistance marker and integration regions of approximately 1000 bp in length and integrates at 303.9° around the D39V chromosome near *spv_1736*. (**D**) Two pPEPZ vectors have either an erythromycin or trimethoprim resistance marker along with an ampicillin marker for selection in *E. coli*. A nonfunctional *E. coli* chromosomal sequence insertion between the *S. pneumoniae* selection marker and the promoter is indicated.

**Figure 2 genes-10-00394-f002:**
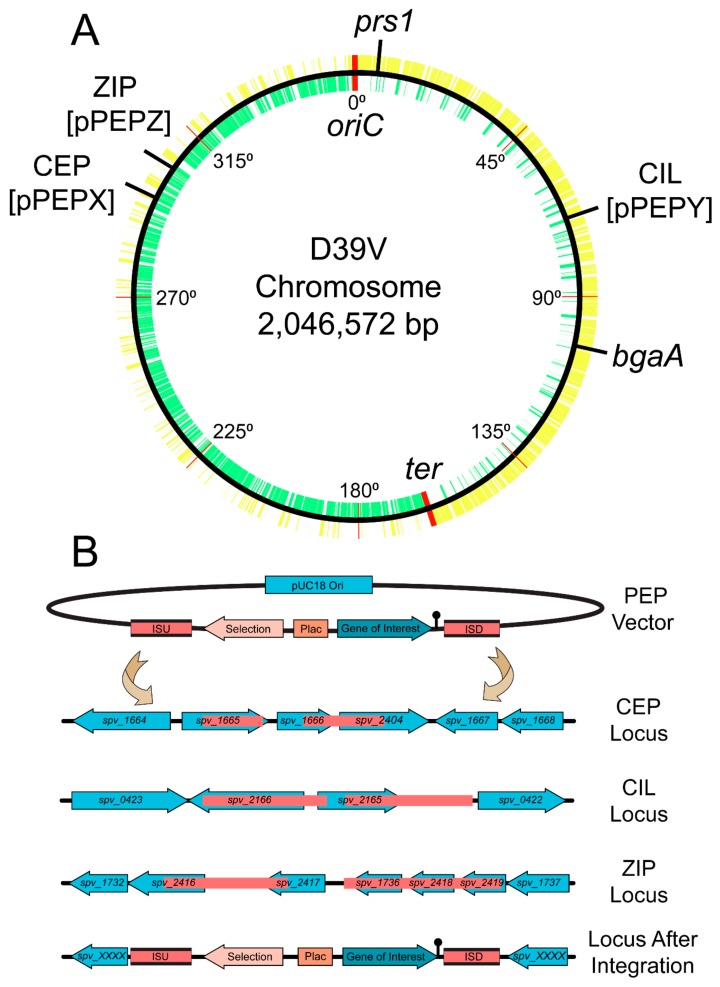
Chromosomal location of integration loci and integration schematic. (**A**) D39V chromosome with all positive strand genes (yellow), negative strand genes (green), and labeled integration loci. Position around the chromosome is demarcated every 45 degrees and origin of replication and terminus are indicated in red. Note that strain D39V has a large inversion around the ter region compared to laboratory strain R6 [14]. (**B**) Representation of insertion sequence upstream (ISU) and insertion sequence downstream (ISD) in pink for available PEP vectors. Double crossover event leads to stable integration in CEP (chromosomal expression platform, pPEPX), CIL (chromosomal integration locus, pPEPY), or ZIP (PEPZ Integration Position, pPEPZ) locus at indicated chromosomal locations based on most recent D39V genome annotation.

**Figure 3 genes-10-00394-f003:**
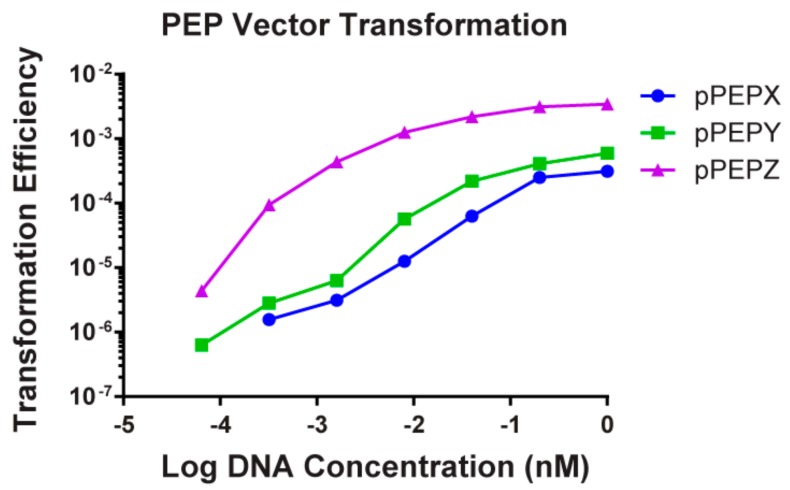
Transformation efficiencies of the various PEP vectors. The three PEP vectors, pPEPX, pPEPY, and pPEPZ were tested for transformation into the *S. pneumoniae* strain D39V. All vectors were transformable into D39V with a dose dependent response based on the amount of DNA added up to a saturation point. At all concentrations of DNA, pPEPZ had the highest transformation efficiency with pPEPX and pPEPY displaying similar rates of transformation and homologous recombination.

**Figure 4 genes-10-00394-f004:**
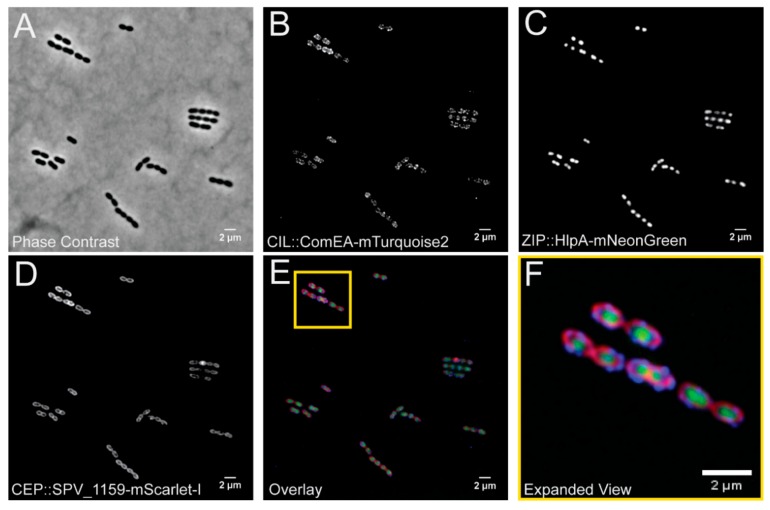
Visualization of a pneumococcal strain VL2706 expressing three different fluorescent fusions simultaneously from the three different integration loci. From the CEP locus (PEPX): ComEA-mTurquoise2; from the CIL locus (PEPY): SPV_1159-mScarlet-I; and from the ZIP locus (PEPZ): HlpA-mNeonGreen. Integration of the three constructed vectors produced distinct localizations based on the proteins used for the fusion. (**A**) A phase contrast image of VL2706. (**B**) The ComEA-mTurquoise2 fusion produced strong foci around midcell as well as at the poles of the cell as previously shown [31], (**C**) HlpA fusions produce distinct green signal that localizes at the chromosome [32,33], and (**D**) SPV_1159 fusions produce distinct membrane associated signals in the bacterial cells. (**E**) An overlay of the three simultaneously expressed fusion proteins and a yellow outline of (**F**) containing an expanded view of the image.

**Figure 5 genes-10-00394-f005:**
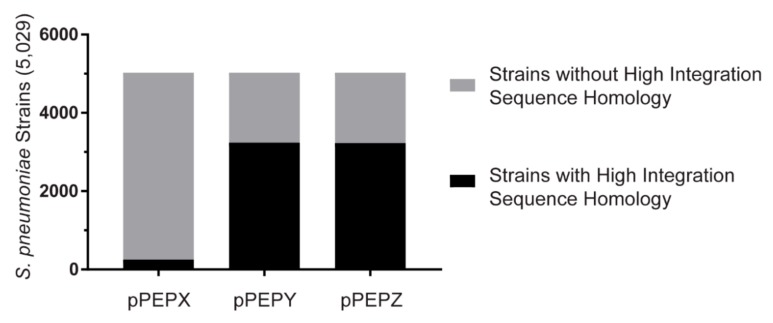
Frequency of *S. pneumoniae* strains with high homology to PEP vector integration regions. Local BLAST database was created containing 5029 genomes of *S. pneumoniae*. Out of these 5029 genomes, 261, 3245, and 3235 strains contained sufficient homology to the pPEPX, pPEPY, and pPEPZ vectors, respectively, to potentially for allow homologous recombination.

**Table 1 genes-10-00394-t001:** List of constructed plasmids. Spec-spectinomycin, Kan-Kanamycin, Gent-Gentamycin, Erm-Erythromycin, Amp-Ampicillin. Tmp-Trimethoprim.

Plasmid Name	Features	Selection-*E. coli*	Selection-*S. pneumoniae*
pPEPX	Promoterless integrative plasmid with terminator	Spec 50 µg/mL	Spec 200 µg/mL
pPEPX-P_lac_	Integrative plasmid with IPTG inducible promoter	Spec 50 µg/mL	Spec 200 µg/mL
pPEPY	Promoterless integrative plasmid with terminator	Kan 50 µg/mL	Kan 250 µg/mL
pPEPY-P_lac_	Integrative plasmid with IPTG inducible promoter	Kan 50 µg/mL	Kan 250 µg/mL
pASR100 (pPEPZ-P_lac_)	Integrative plasmid with IPTG inducible promoter	Spec 50 µg/mL	Spec 200 µg/mL
pASR102 (pPEPZ-P_lac_)	Integrative plasmid with IPTG inducible promoter	Gent 40 µg/mL	Gent 50 µg/mL
pASR103 (pPEPZ-P_lac_)	Integrative plasmid with IPTG inducible promoter (Amp selection for *E. coli*)	Amp 100 µg/mL	Erm 0.5 µg/mL
pASR130 (pPEPZ-P_lac_)	Integrative plasmid with IPTG inducible promoter (Amp selection for *E. coli*)	Amp 100 µg/mL	Tmp 10 µg/mL
pPEPY-P_lac_-Link-mTurquoise2	Integrative plasmid with IPTG inducible promoter and linker for C-terminal fusion of mTurquoise2	Kan 50 µg/mL	Kan 250 µg/mL
pPEPY-P_lac_-Link-mNeonGreen	Integrative plasmid with IPTG inducible promoter and linker for C-terminal fusion of mNeonGreen	Kan 50 µg/mL	Kan 250 µg/mL
pPEPY-P_lac_-Link-mScarlet-I	Integrative plasmid with IPTG inducible promoter and linker for C-terminal fusion of mScarlet-I	Kan 50 µg/mL	Kan 250 µg/mL

**Table 2 genes-10-00394-t002:** Transformation efficiency of PEP vectors into various pneumococcal strains. Spec—spectinomycin, Kan—Kanamycin, Gent—Gentamycin, Erm—Erythromycin, Tmp—Trimethoprim. *—pPEPZ plasmid chosen based on susceptibility of strain.

Strain	Serotype	Susceptibility	PEP Transformable Based on Sequence	Transformation Efficiency (pPEPX)	Transformation Efficiency (pPEPY)	Transformation Efficiency (pPEPZ *)
PMEN1	23F	Erm, Spec, Kan	pPEPY and pPEPZ	2.56 × 10^−8^	2.61 × 10^−7^	8.17 × 10^−6^
PMEN3	9V	Erm, Spec, Kan	pPEPZ	2.49 × 10^−8^	3.31 × 10^−7^	2.26 × 10^−8^
PMEN14	19F	Gent, Spec, Kan	pPEPZ	8.10 × 10^−7^	8.52 × 10^−7^	1.43 × 10^−6^
PMEN18	14	Gent, Spec, Kan	pPEPY and pPEPZ	3.77 × 10^−7^	0	2.16 × 10^−6^
PMEN28	1	Erm, Gent, Spec, Kan	N/D	8.79 × 10^−8^	7.95 × 10^−8^	1.03 × 10^−6^
TIGR4	4	Erm, Gent, Spec, Kan, Tmp	pPEPX and pPEPZ	4.98 × 10^−6^	0	2.42 × 10^−5^

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
