# Peer review of "Three New Integration Vectors and Fluorescent Proteins for Use in the Opportunistic Human Pathogen Streptococcus pneumoniae"

_genes, 2019, doi:10.3390/genes10050394_

Round 1
Reviewer 1 Report
The manuscript by Keller et al describes the construction and use of several integrative cloning vectors for use in S. pneumoniae. The authors have used their existing RNAseq database to identify areas that are transcriptionally silent and relatively well conserved within the species to serve as areas for gene insertion. The vectors contain a variety of resistance genes that would allow for multiple separate insertions in a single background and use in more resistant strains. The vectors are created with an inducible lac promoter allowing for controlled gene expression after insertion of an already described lacR repressor. In addition, the authors introduce three new fluorescent proteins for use in S. pneumoniae and demonstrate their utility in protein localization studies. Although the work is purely tool development, the plasmid content was carefully considered and the utility of the plasmids clearly demonstrated. In addition, the validation that plasmids can be used in many different strain backgrounds expands the utility significantly. The constructed plasmids will be a valuable addition to the pneumococcal research community. The authors plan to make these tools publically available through Addgene which should make acquisition by the public straightforward.
The manuscript is clearly written and the properties of each plasmid clearly depicted in figures and text (with a few minor exceptions, noted below).
I have a few minor comments:
1. Please check for proper italicization of S. pneumoniae and E. coli throughout (line 267, 278, 358, etc).
2. The authors use circular plasmids and do not describe how they ensured that a double cross over event occurred after transformation (rather than a single crossover resulting in a duplication and retention of the plasmid backbone). This would seem to be an important consideration, especially if several vectors are used in a single strain because of shared sequences between plasmids in this area. It is also an important consideration when the E. coli plasmid backbone carries ampicillin resistance, which is not a resistance marker that should be carried in S.pneumoniae. Based on the figures supplied in supplemental data, there are some strain backgrounds where a single cross over may occur preferentially over a double crossover because of homology with only one side of the insertion point. This potential should be discussed.
3. The legends for Figures 1 and 2 are swapped.
4. I believe that the differences between the plasmids would be clearer if, in Figure 2, the actual integrative locus designations were used instead of “ISU” and “ISD” since this is the major difference between the plasmids.
5. It is unclear where transcriptional terminators are cloned. This confusion arised from multiple places in the manuscript:
Line 137-8 “This insertion 137 of 777 bp is from E. coli chromosomal DNA and does not affect the functioning of the plasmid, as the 138 Plac promoter is still well insulated by the transcriptional terminators…” If there are terminators before the Plac promoter preventing read through from the upstream fragment, they are not depicted in the diagrams in figure 2. The only ter sequence depicted follows the MCS.
Similarly, in the construction of pPEPY (line 97) there is mention of three terminators being added, but it is not clear where these were placed based on the descriptions provided.
Also wording (line 242-3) such as “Transcriptional terminators flank the MCS in all vectors to ensure that the inserted DNA is not influenced by transcriptional events happening elsewhere on the chromosome…” Of course the terminators must not flank the MCS on both sides or the Plac promoter wouldn’t work. The description and depiction of terminators needs to be clarified.
6. Was the growth curve depicted in Supplemental Figure 1 done with IPTG? I’m not sure if the point here was to show that simple integration into the three sites didn’t affect growth or if both integration and expression don’t affect growth. It seems that the former is probably more meaningful in terms of the utility of these insertion plasmids. If the point was to show that the triple integration of foreign DNA into the three sites doesn’t affect growth of the organism (independent of expression of the inserted genes) then I would remove the word “expressing” (line 441) from the legend and clarify this in the text. Also, either a figure panel or a line of text in reference to figure 4 that non-induced cells do not fluoresce (I assume this is true).
7. Line 278 – “These data are…”
8. Line 352 – change “have high sequence” to “both having high sequence” or “the fact that both have…” or similar.
Author Response
Reviewer One responses
Major Reviewer Comment
1. If the authors wish the claim a high site-specific integration of these vectors, they need to provide some evidence that the vectors consistently integrate at the expected site (at least within D39V) or the rate in which they ‘mis-target’ and integrate elsewhere. They have included portions of the pneumococcal chromosome into the vectors to direct site-specific homologous recombination, but have not demonstrated if this provides enough specificity that the vectors always integrate at the expected site (and not elsewhere in the chromosome). When testing these vectors in clinical isolates, they state “Even when we did not find a high level of homology [referring to the integration site], we could still transform strains PMEN3 and PMEN14 with plasmids pPEPX and pPEPY”. I therefore think it is important the authors demonstrate some evidence of the integration specificity of these vectors e.g. by whole genome sequencing or PCR screening or Southern blot hybridisation of transformants. I think at least ~10 transformants (D39V background) of each plasmid should be tested to determine where the vector has integrated in each transformant.
Response: Thank you for the comment and we agree with your concern. Primers necessary for checking double cross over integration were added in Ln 196-197. Also text was added about the efficiency of correct double crossover integration (Ln286-291) as well as addition of a new supplementary figure demonstrating double crossover through PCR.
Minor Reviewer Comment
1. I believe the figure legends for Fig 1 and 2 are the wrong way around (or at least presented in the wrong order compared to the order the figures are placed in the manuscript)?
Response: You are correct, the legends were inverted.
2. Line 48: should be “decade” not “decades”?
Response: Corrected
3. To further assist future researchers in using these vectors, it would be appreciated if authors can include concentrations for antibiotics used to maintain the vectors in E. coli (and even the concentrations used for selecting for pneumococcal transformants, although this might vary by strain, it would be useful for researchers to have a starting point that is known to work for D39V).
Response: This would be a useful addition to the paper. We have added an extra column to Table 1 adding this information.
4. Restriction site nomenclature – only the part of the enzyme name that comes from the species is italicised (not the strain name). e.g. “Eco” should be italicised as that is named after the species, “RI” should not as that is the strain from which the enzyme was isolated.
Response: Thank you for this clarification, appropriate changes have been made.
5. Line 117: “gentamicin” is used here but “gentamycin” is used elsewhere in the manuscript
Response: Corrected
6. Line 261: “the strain” – I assume the authors are referring to pneumococcal strain here?
Response: Clarification was added to the sentence.

Reviewer 2 Report
In this manuscript, Keller and colleagues describe the construction and applicability of a series of suicide vectors for genetic manipulation of Streptococcus pneumoniae, a human pathogen of global importance. This includes 3 vectors that differ in integration site and selectable marker, making multiple genetic manipulations within the same strain possible, as well as compatibility of multiple cloning sites to allow robust sub-cloning across the different vectors. These features (in addition to others mentioned throughout the manuscript) make these vectors a powerful tool for use in the pneumococcal field.
In particular, the authors should be praised for the thought and considerations taken into account in constructing these vectors such as utilizing RNA-Seq data to identify transcriptionally silent regions as integration sites (ensuring minimal disturbance to the bacterial host and reducing the chance of indirectly inducing phenotypic changes) and testing these vectors in other strains to demonstrate the applicability of these vectors in genetically modifying laboratory and clinical strains.
I have one comment (and a few minor suggestions) that I would like the authors to address:
1. If the authors wish the claim a high site-specific integration of these vectors, they need to provide some evidence that the vectors consistently integrate at the expected site (at least within D39V) or the rate in which they ‘mis-target’ and integrate elsewhere. They have included portions of the pneumococcal chromosome into the vectors to direct site-specific homologous recombination, but have not demonstrated if this provides enough specificity that the vectors always integrate at the expected site (and not elsewhere in the chromosome). When testing these vectors in clinical isolates, they state “Even when we did not find a high level of homology [referring to the integration site], we could still transform strains PMEN3 and PMEN14 with plasmids pPEPX and pPEPY”. I therefore think it is important the authors demonstrate some evidence of the integration specificity of these vectors e.g. by whole genome sequencing or PCR screening or Southern blot hybridisation of transformants. I think at least ~10 transformants (D39V background) of each plasmid should be tested to determine where the vector has integrated in each transformant.
2. Minor comments:
· I believe the figure legends for Fig 1 and 2 are the wrong way around (or at least presented in the wrong order compared to the order the figures are placed in the manuscript)?
· Line 48: should be “decade” not “decades”?
· To further assist future researchers in using these vectors, it would be appreciated if authors can include concentrations for antibiotics used to maintain the vectors in E. coli (and even the concentrations used for selecting for pneumococcal transformants, although this might vary by strain, it would be useful for researchers to have a starting point that is known to work for D39V).
· Restriction site nomenclature – only the part of the enzyme name that comes from the species is italicised (not the strain name). e.g. “Eco” should be italicised as that is named after the species, “RI” should not as that is the strain from which the enzyme was isolated.
· Line 117: “gentamicin” is used here but “gentamycin” is used elsewhere in the manuscript
· Line 261: “the strain” – I assume the authors are referring to pneumococcal strain here?
Author Response
Reviewer Two responses
Reviewer Comments
1. Please check for proper italicization of S. pneumoniae and E. coli throughout (line 267, 278, 358, etc).
Response: Thank you for the careful reading and this issue has been addressed.
2. The authors use circular plasmids and do not describe how they ensured that a double cross over event occurred after transformation (rather than a single crossover resulting in a duplication and retention of the plasmid backbone). This would seem to be an important consideration, especially if several vectors are used in a single strain because of shared sequences between plasmids in this area. It is also an important consideration when the E. coli plasmid backbone carries ampicillin resistance, which is not a resistance marker that should be carried in S. pneumoniae. Based on the figures supplied in supplemental data, there are some strain backgrounds where a single cross over may occur preferentially over a double crossover because of homology with only one side of the insertion point. This potential should be discussed.
Response: Thank you for making this point, we agree that this is an important consideration that we overlooked in the manuscript preparation. Primers necessary for checking double cross over integration were added in Ln 196-197. Also text was added about the efficiency of correct double crossover integration (Ln286-291) as well as addition of a new supplementary figure demonstrating double crossover through PCR.
3. The legends for Figures 1 and 2 are swapped.
Response: This has been corrected
4. I believe that the differences between the plasmids would be clearer if, in Figure 2, the actual integrative locus designations were used instead of “ISU” and “ISD” since this is the major difference between the plasmids.
Response: We agree that this can help with clarification of differences between the plasmids. We altered the figure by keeping the “ISU” and “ISD” nomenclature, but indicated the gene where integration occurred in parenthesis.
5. It is unclear where transcriptional terminators are cloned. This confusion arised from multiple places in the manuscript:
Line 137-8 “This insertion 137 of 777 bp is from E. coli chromosomal DNA and does not affect the functioning of the plasmid, as the 138 Plac promoter is still well insulated by the transcriptional terminators…” If there are terminators before the Plac promoter preventing read through from the upstream fragment, they are not depicted in the diagrams in figure 2. The only ter sequence depicted follows the MCS.
Response: Thank you for indicating this, the figure has been modified with the addition of an additional indicator of the terminator before the promoter as well as adding the “TER” nomenclature in the figure.
Similarly, in the construction of pPEPY (line 97) there is mention of three terminators being added, but it is not clear where these were placed based on the descriptions provided.
Response: Additional clarification was added to indicate that these terminators are added after the MCS (Ln 98) and are present in the cloning site amplified from pPEPX (Ln 94-95)
Also wording (line 242-3) such as “Transcriptional terminators flank the MCS in all vectors to ensure that the inserted DNA is not influenced by transcriptional events happening elsewhere on the chromosome…” Of course the terminators must not flank the MCS on both sides or the Plac promoter wouldn’t work. The description and depiction of terminators needs to be clarified.
Response: We agree that clarification is needed and have altered the text to indicate that terminators flank the promoter and MCS, not just the MCS (Ln 247).
6. Was the growth curve depicted in Supplemental Figure 1 done with IPTG? I’m not sure if the point here was to show that simple integration into the three sites didn’t affect growth or if both integration and expression don’t affect growth. It seems that the former is probably more meaningful in terms of the utility of these insertion plasmids. If the point was to show that the triple integration of foreign DNA into the three sites doesn’t affect growth of the organism (independent of expression of the inserted genes) then I would remove the word “expressing” (line 441) from the legend and clarify this in the text. Also, either a figure panel or a line of text in reference to figure 4 that non-induced cells do not fluoresce (I assume this is true).
Response: Supplementary Figure 1 (Now supplementary figure 2) is supposed to indicated that integration of the plasmids do not effect growth independent of expressing any genes contained within the plasmids. Text has been altered for clarity (Ln317-318). The text of the supplementary figure legend has also been altered (Ln464) and text was added to indicated fluorescence is seen upon induction with IPTG (Ln314).
7. Line 278 – “These data are…”
Response: Corrected
8. Line 352 – change “have high sequence” to “both having high sequence” or “the fact that both have…” or similar.
Response: Corrected
